

# Conventional single-gap $s$−wave superconductivity and hidden peak effect in single crystals of $Mo_8Ga_{41}$ superconductor

Sunil Ghimire[1,2], Kyuil Cho[1,2∘], Kamal R. Joshi[1,2], Makariy A. Tanatar[1,2],
Zhixiang Hu[3,4], Cedomir Petrovic[3,4,5,6] and Ruslan Prozorov[1,2⋆]

**1** Ames National Laboratory, Ames, Iowa 50011, U.S.A.
**2** Department of Physics & Astronomy, Iowa State University, Ames, Iowa 50011, U.S.A.
**3** Condensed Matter Physics and Materials Science Department,
Brookhaven National Laboratory, Upton, New York 11973, U.S.A.
**4** Department of Physics and Astronomy, Stony Brook University,
Stony Brook, New York 11794-3800, U.S.A.
**5** Shanghai Key Laboratory of Material Frontiers Research in Extreme Environments (MFree),
Shanghai Advanced Research in Physical Sciences (SHARPS),
Pudong, Shanghai 201203, China
**6** Department of Nuclear and Plasma Physics, Vinca Institute of Nuclear Sciences,
University of Belgrade, Belgrade 11001, Serbia

⋆ prozorov@ameslab.gov

## Abstract

London and Campbell penetration depths were measured in single crystals of the endo-hedral gallide cluster superconductor, $Mo_8Ga_{41}$. The full temperature range superfluid density, $\rho_s(T)$, is consistent with the clean isotropic $s$−wave weak-coupling BCS theory without any signs of the second gap or strong coupling. The temperature dependence of the Campbell length is hysteretic between zero-field cooling (ZFC) and field-cooling (FC) protocols, indicating an anharmonic vortex pinning potential. The field dependence of the effective critical current density, $j_c(H)$, reveals an unusual result. While in the ZFC protocol, $j_c(H)$ is monotonically suppressed by the magnetic field, it exhibits a profound "hidden" peak effect in the FC protocol, that is, without a vortex density gradient. We suggest a possible novel mechanism for such a peak effect, which involves both static and dynamic aspects of vortex pinning.

## Contents

∘Current affiliation: Department of Physics, Hope College, Holland, MI 49423, U.S.A.

## 1 Introduction

Unconventional superconductivity is a perpetual theme in current research. Particular attention is devoted to multigap superconductivity [1,2]. Although it was theoretically introduced in 1959 [3,4] and later received experimental confirmation [5], it has not become a part of the mainstream research effort. All has changed with the discovery of two-gap superconductivity in $MgB_2$ in 2001 [6], after which multigap physics has become one of the most studied phenomena in modern superconductivity [1,2]. Inspired by $MgB_2$, many binary systems have been investigated, and endohedral gallide cluster compounds are among the actively researched families of materials [7–10].

In this paper, we study one of its members, $Mo_8Ga_{41}$, with superconducting transition temperature $T_c = 9.7$ K and an upper critical field $H_{c2} \approx 8.3 - 8.7$ T [7,10,11]. Although this compound has been known for more than forty years [7], it has attracted recent attention for possible deviations from the conventional single-gap $s-$wave weak-coupling Bardeen-Cooper-Schrieffer (BCS) superconductivity [12,13]. However, the experimental situation is quite complicated.

A quantum oscillations study of $Mo_8Ga_{41}$ inferred three-dimensional electronic bands with strong coupling to phonons [14]. Similarly, a $T-$linear resistivity observed in a wide temperature interval was interpreted to be due to scattering from low-lying phonon modes, which could lead to enhanced electron-phonon coupling [15]. A scanning tunneling microscopy (STM) study of $Mo_8Ga_{41}$ crystals suggested weak-coupling superconductivity with two gaps of 1.6 meV and 0.9 meV, yielding the ratios $\Delta(0)/k_B T_c$, of 2.1 and 1.2, and concluded that this behavior is similar to $MgB_2$ [9]. A combined study using ac calorimetry and scanning tunneling microscopy (STM) on the same high-quality crystals showed only one intrinsic gap and some traces of additional superconducting phases, which could be detected as the second gap in the STM measurements [16].

Several studies reported measurements of the heat capacity and superfluid density. However, despite significant effort, so far the results cannot be reconciled within any single model. Specifically, heat capacity measurements show a normalized jump at the superconducting transition $\Delta C/C(T_c) = 2.83$ [17] and close to 3.0 [18], twice the value of 1.43 predicted by weak-coupling BCS theory [19] (the original BCS paper gives the value of 1.52 [13]). The same groups reported measurements related to the superfluid density. The results of the muon spin resonance ($\mu$SR) measurements are consistent with a single-gap model with somewhat enhanced $\Delta(0)/k_B T_c = 2.1$ [20] compared to the weak-coupling BCS value of 1.76 [12,13]. However, to reconcile with their specific heat study, the same group plotted their $\mu$SR superfluid density with a two-gap model with two very large gaps, 4.3 meV and 1.76 meV, yielding the gap ratios $\Delta(0)/k_B T_c =$5.1 and 2.1, respectively [17]. Similarly, the lower critical field, $H_{c1}(T)$, (in lieu of the superfluid density) was fitted with expressions close to (and even somewhat smaller than) the weak-coupling values, but very far from the theoretical $H_{c1}(T)$ calculated with the ratio $\Delta(0)/k_B T_c = 2.2$ estimated from the specific heat data collected by the same group [18]. We note that unusually large specific heat jump at $T_c$ is observed in other low$-T_c$ systems, for example, $Rh_{17}S_{15}$ [21], $CeCoIn_5$, $CeRhIn_5$, $U_6Fe$, $UBe_{13}$, $PrOs_4Sb_{12}$, $NpPd_2Al_2$ (see Ref. [22] for review). The origin of these observations is not understood and requires further investigation.

On a general note, while the superfluid density is the quantity directly related to the super-conducting gap structure [23], total specific heat contains contributions from all excitations in the system and it is not trivial to separate its zero-field electronic part considering substantial magnetoresistance [15]. Still, assuming that the electronic specific heat was estimated correctly, for the data analysis both groups used non-self-consistent empirical models that are inapplicable for a number of reasons: (1) both groups used dirty limit expressions, whereas $Mo_8Ga_{41}$ appears to be rather clean, as is evident from the residual resistivity ratio of 15.4 [16] and, importantly, the observation of quantum oscillations [14]. (Our results are also consistent with the clean limit). (2) The use of approximate weak-coupling temperature-dependent gaps for the fitting while analyzing the strong-coupling regime. The proper self-consistent strong-coupling analysis using Eliashberg theory shows that even in the case of an isotropic single $s-$ wave gap, the order parameter temperature dependence is quite different from the weak-coupling BCS [24, 25]. (3) The two-gap situation, suggested by their analysis, is even worse, because the self-consistent two-gap solution produces gaps with different temperature dependencies, which are different from a single-gap BCS for both weak [26] and strong [27, 28] coupling limits. Not surprisingly, the temperature-dependent heat capacity could not be fitted with a two-gap model, see Fig.4 of Ref. [18]. With regard to superfluid density, the suggested ratio $\Delta(0)/k_B T_c = 2.2$ yields the strong coupling parameter $T_c/\omega_{ln} \approx 0.1$ [24]. This results in a superfluid density quite different from the weak-coupling one, as shown by the dotted line in Fig.111 of Ref. [24]. Again, the two-gap situation would only exacerbate the difference.

In summary, there is a significant disagreement about the superconducting gap structure of $Mo_8Ga_{41}$ from the thermodynamic measurements. Further investigation is needed and in this contribution we show that $Mo_8Ga_{41}$ is a conventional single-gap $s-$wave BCS superconductor.

With regard to vortex properties, there is only very limited literature. Magnetic measurements of polycrystalline samples showed conventional-looking hysteresis $M(H)$ loops with the specific power-law magnetic field dependence of irreversible magnetization, indicative of a strong pinning [29–34]. The persistent current density in zero field was estimated from the Bean model [35, 36], $j_p(T = 2\,K) = 0.3\,MA/cm^2$ [37]. Another study of single crystals using miniature Hall probe arrays estimated $j_p(T = 2\,K) = 0.016\,MA/cm^2$ from the Maxwell equation [38]. In principle, such a difference can be attributed to the dissimilarity between the poly-crystalline and single-crystalline samples. The latter study confirmed the strong-pinning scenario and suggested single-gap superconductivity, perhaps contaminated by secondary phases.

One of the most interesting mixed-state features is the non-monotonic dependence of the irreversible component of magnetization on a magnetic field or temperature. Depending on the context and author's preferences, this feature can be called the "peak effect", the "second magnetization peak" or the "fishtail" [39–50]. Since any measurement has a certain experimental time window, the measured magnetic moment and hence the persistent current density $j_p$, are affected by magnetic relaxation, which is exponentially fast at current densities close to the critical current $j_c > j_p$ [44, 45, 51]. Consequently, there is an ongoing debate about the static or dynamic origin of the peak effect. The "static" explanation suggests actual non-monotonic behavior of the unrelaxed critical current, $j_c(H)$, which would imply an unusual pinning mechanism, for example, due to the softening of the vortex lattice at low fields and close to $H_{c2}$ [40, 52], two different vortex phases [53], or a crossover from collective to plastic creep mechanism [47, 54]. The "dynamic" explanation involves a field-dependent magnetic relaxation that is faster at low magnetic fields, for example, in the weak collective pinning and creep model [44].

In this work, we address both the superconducting gap structure probed by measuring the London penetration depth and the theoretical critical current density probed by measuring the Campbell penetration depth [33, 55–57] in $Mo_8Ga_{41}$ single crystals. Although the superfluid density, $\rho_s(T)$, is well described by the isotropic single-gap weak-coupling BCS theory, the

vortex behavior is unusual. We found an unexpected "hidden" peak effect in the "true" $j_c(H)$ in the field-cooling protocol, when persistent current density is zero (no vortex density gradient) which is inaccessible to other types of magnetization measurements, global or local. We note that from many superconductors in which we measured Campbell length, there is only one example, LiFeAs, where we observed similar behavior, namely, monotonic in field ZFC current density and non-monotonic peak effect in a FC protocol [58].

## 2  Samples and methods

**Samples:**  Single crystals of $Mo_8Ga_{41}$ were grown by the high-temperature self-flux method. Mo and Ga were mixed in an 8 : 500 ratio in an alumina crucible and sealed in an evacuated quartz tube. The ampule was heated up to $850^0$C in two hours, held at $850^0$C for 10 hours, and then slowly cooled to $170^0$C for 55 hours, when the crystals were decanted. The details of crystal growth and characterization are given in Ref. [14].

**Lower critical field:**  The lower critical field, $H_{c1}$, was measured using local optical magnetometry based on nitrogen vacancy centers in diamond (NV$^-$centers) [59–61]. Due to the specifics of the NV-center low-energy spectrum, the stimulated fluorescence amplitude depends on the applied magnetic due to Zeeman splitting. If the magnetic field is applied along the $\hat{z}$ direction, this results in two peaks in the optically detected magnetic resonance (ODMR) with their splitting, $\Delta f = 2g_{NV}B/\sqrt{3}$, where $g_{NV} = 2.8$ MHz/Oe is the NV center gyromagnetic ratio and $\sqrt{3}$ in the denominator takes into account the possible projections of the applied magnetic field on the $N - V$ bond direction in our [100] oriented diamond crystalline film. The NV centers are implanted 20 nm below the surface and the diamond film is placed on top of the flat sample. Measurements are performed as close to the edge of the sample, approximately 10 $\mu m$. This type of measurement is similar to the micro-Hall probe measurements mentioned in the Introduction [18].

**London penetration depth:**  The London penetration depth, $\lambda_L(T)$, was measured using a sensitive frequency-domain self-oscillating tunnel-diode resonator (TDR) operating at a frequency around 14 MHz. The measurements were performed in a $^3$He cryostat with the base temperature of about $T_{min} = 0.4$ K, which is $0.041T_c$, which gives us enough range to examine the low-temperature limit, which starts below $T_c/3$, where the superconducting gap is approximately constant. The experimental setup, measurement protocols, and calibration are described in detail elsewhere [62–65].

In the experiment, the temperature dependence of the relative resonant frequency shift $\delta f = f(T) - f(T_{min})$ is measured. This quantity is proportional to the magnetic susceptibility of the sample, which, in turn, is proportional to $\delta f/f_0 = G\Delta\lambda(T)/R$ where $R$ is the effective dimension of the sample calculated for each particular geometry and $G$ is the dimensionless calibration constant [65]. The parameter $f_0 = 14$ MHz is the frequency of the empty resonator. The calibration constant $G$ is measured by physically pulling the sample out of the coil at $T_{min}$. The dimensions of the sample used in this experiment was $400 \times 200 \times 190\,\mu m$ and its effective is $R = 44.53\,\mu m$, determined from the established calibration procedure [65]. The total London penetration depth is obtained as $\lambda(T) = \lambda(0) + \Delta\lambda(T)$, where $\Delta\lambda(T) = \lambda(T) - \lambda(0.4\,K)$ is the measured change in the penetration depth as described above and the absolute value $\lambda(0)$ is estimated separately using the NV-centers optical magnetometry. Finally, the normalized superfluid density that can be directly compared with the theory is evaluated as $\rho_s(T) = (\lambda(0)/\lambda(T))^2$.

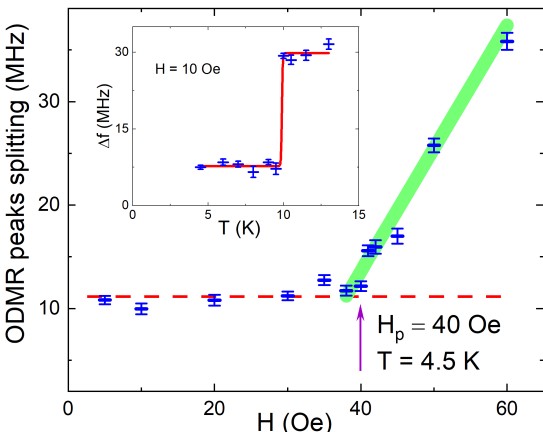

Figure 1: Determining the magnetic field of the first flux penetration, $H_p$, in $Mo_8Ga_{41}$ at $T = 4.5$ K by local optical magnetometry using NV-centers. Zeeman splitting is measured at the edge of the superconductor after zero field cooling. A sharp departure from a constant value occurs at $H_p = 40$ Oe when vortices start penetrating the sample. The inset shows sharp superconducting transition measured at the same spot.

**Campbell penetration depths:** The Campbell penetration depth, $\lambda_C(T)$, is measured exactly the same way as the London penetration depth, but in a finite applied magnetic field, which produces Abrikosov vortices in the sample. Then, the measured penetration depth, $\lambda_m$, has two contributions, the usual London penetration depth, $\lambda$, and the Campbell penetration depth $\lambda_C$, which is a characteristic length scale over which a small ac perturbation is transmitted elastically by the vortex lattice into the sample [31, 32, 55, 56, 66]. More specifically, the amplitude of the ac perturbation must be small enough so that the vortices remain in their potential well, and their motion is described by the reversible linear elastic response. In this case, $\lambda_m^2 = \lambda^2 + \lambda_C^2$ [57, 67]. This requirement of a very small amplitude makes most conventional ac susceptibility techniques inapplicable to Campbell length measurements. In our case, the excitation magnetic field is approximately 20 mOe, which is surely well below $H_{c1}$ for most of the temperature range. It is important to note that conventional ac and dc measurements, where displacement of vortices out of their potential wells is involved, probe the Bean persistent current density [44, 45], whereas Campbell length measurements probe the curvature of the effective pinning potential [31, 32, 67]. This information is inaccessible for conventional measurements.

## 3 Results and discussion

### 3.1 The absolute value of $\lambda_L(0)$ from NV-centers optical magnetometry

Figure 1 shows the ODMR splitting near the edge of the sample as a function of the applied dc magnetic field at $T = 4.5$ K. A sharp break from a constant value occurs in the magnetic field of the first flux penetration $H_p(4.5 \, \text{K}) = 40$ Oe. The inset shows a superconducting transition measured at the same spot. Using a revised effective demagnetization factor for a $2a \times 2b \times 2c$ cuboid, $N^{-1} = 1 + \frac{3c}{4a}(1 + \frac{a}{b})$ [68], the true $H_{c1}(4.5 \, \text{K}) = 85$ Oe is obtained. The absolute value of the London penetration depth is then estimated by solving $H_{c1} = \frac{\phi_0}{4\pi\lambda^2}\left(\ln\frac{\lambda}{\xi} + 0.497\right)$ [69], where $\xi = \sqrt{\phi_0/2\pi H_{c2}}$ is the coherence length and $\phi_0$ is the magnetic flux quantum. The upper critical field at the temperature of interest, $H_{c2}(4.5 \, \text{K}) = 5.25$ T, was obtained from

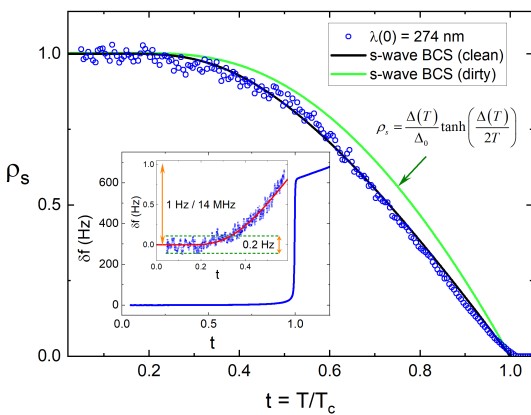

Figure 2: Main panel: superfluid density, $\rho_s(T) \equiv \lambda^2(0)/\lambda^2(T) = (1 + \Delta\lambda(T)/\lambda(0))^{-2}$ calculated using the absolute value of London penetration depth, $\lambda(0) = 274$ nm, and the temperature-dependent variation, $\Delta\lambda(T)$ measured using a tunnel diode resonator. The black line shows a theoretical curve (not a fit!) for the isotropic $s-$wave weak-coupling BCS superconductor. The green curve shows the numerical dirty limit, well approximated analytically by $\rho_s = (\Delta(T)/\Delta(0)) \tanh(\Delta(T)/2T)$. The outer inset shows the raw data, $\delta f(t)$, in the full temperature range. The inner inset shows low-temperature zoom with the data shown by blue circles and low-temperature BCS asymptotic shown by the red line.

the specific heat and magnetization measurements of high-quality crystals [17], which gives $\xi(4.5\,\mathrm{K}) = 8$ nm. From this we obtain $\lambda(4.5\,\mathrm{K}) = 281$ nm. Finally, using a known analytic approximation of $\lambda(T)$ for an $s-$wave weak-coupling superconductor [70], we can evaluate $\lambda(0) = \lambda(t)\sqrt{1-t^4}$, resulting in $\lambda(0) = 274$ nm. This value will be used to construct the superfluid density.

## 3.2 London penetration depth and superfluid density, $\rho_s(T)$

We are now ready to calculate the normalized superfluid density,

$$\rho_s(T) \equiv \lambda^2(0)/\lambda^2(T) = (1 + \Delta\lambda(T)/\lambda(0))^{-2}.$$

The result is shown in Fig.2. The raw data, the resonator frequency shift, $\delta f(T)$, in a full temperature range are shown in the inset and the low temperature range is shown in the inner inset. We present the raw data to demonstrate that some apparent noise is not due to measurement problems but to a very small signal. The inner inset of Fig.2 shows that the total frequency change when the temperature increases from $T_{min}$ to $0.5T_c$ is only 1 Hz. The noise level indicated by green dashed lines is 0.2 Hz, which means that we have a 10 parts per billion accuracy, which is very good. The red line in the inner inset is the isotropic asymptotic, $\delta f(T) = A\sqrt{\pi\delta/2t}e^{-\delta/t}$ with a fixed ratio $\delta = \Delta(0)/k_BT_c \approx 1.76$ leaving only one free scaling parameter, $A$. This equation is applicable to $\delta f$, because, as described in Section 2, $\delta f \sim \Delta\lambda$.

Next we calculated the normalized superfluid density using $\lambda(0) = 274$ nm determined using NV optical magnetometry as described in Section 3.1. The main panel of Fig.2 shows the data (blue circles) and two fixed curves (not a fit!) for clean (black) and dirty (green) limits of the isotropic $s-$wave weak-coupling BCS superconductor. The non-magnetic dirty limit has an analytic form, $\rho_s = (\Delta(T)/\Delta(0)) \tanh(\Delta(T)/2T)$ [71], and does not fit our results. The theoretical curve in the clean limit was calculated numerically using a self-consistent Eilenberger theory [26,72]. Clearly, the clean limit fits the very data well in the entire temperature range. As mentioned in the Introduction, the clean limit is independently supported by observations of quantum oscillations in $Mo_8Ga_{41}$ crystals [14] and a not too low residual resistivity ratio of 15.4 [16].

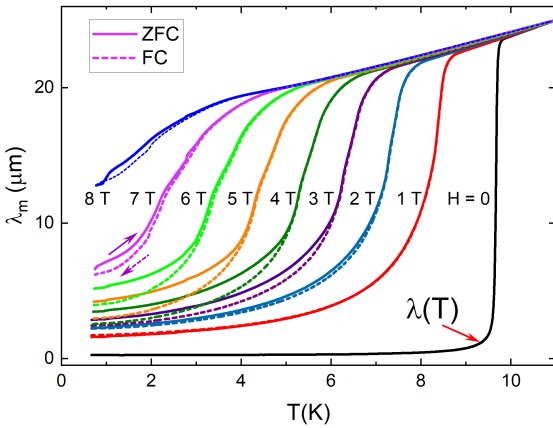

Figure 3: Total magnetic penetration depth, $\lambda_m(T) = \lambda(0) + \Delta\lambda(T)$, as a function of temperature, measured on warming (ZFC, solid lines) and cooling (FC, dashed lines) at different applied dc magnetic fields indicated next to each curve. Two arrows near a 7 T curve indicate the direction of the temperature sweep.

Addressing the possibility of two distinct gaps, we note that, unlike the not-self-consistent $\alpha-$model where temperature dependence of the two gaps is assumed to be BCS-like, the self-consistent two-band treatment shows that the smaller gap becomes significantly different from what is expected from the BCS [26]. If the interband pairing is not too strong, the resulting superfluid density, $\rho_s(T)$, develops a visible suppression at intermediate and higher temperatures. (In the limit of zero interband pairing, there will be two different $T_c$ values!). None of the reported measurements so-far showed such features in the temperature dependencies of the gaps or the superfluid density. The tunneling study suggested that their results are similar to MgB$_2$ [9]. However, this is not the case from the superfluid density point of view because in MgB$_2$, $\rho_s(T)$ is clearly suppressed at elevated temperatures, and this was one of the signatures that helped to identify two-gap physics [73,74].

### 3.3 Campbell penetration depth and critical current density

Figure 3 shows the total magnetic penetration depth, $\lambda_m$, measured upon warming (solid curves) after zero-field cooling (ZFC, dashed curves)and on cooling from above $T_c$ (FC) at different applied magnetic fields, shown in the figure. The hysteretic behavior between ZFC and FC protocols is not due to vortex density gradient as occurs in dc magnetization measurements. This hysteresis comes from a non-parabolic vortex pinning potential, $U(r)$. Note that the measured $\lambda_m(T)$ does not diverge approaching the normal state. Of course, the actual penetration depth diverges at $T \to T_c(H)$, but in the normal state, the measured depth $\lambda_m$ cannot exceed the skin depth, $\delta_{\text{skin}} = \sqrt{\rho/\mu_0\pi f}$, where $\mu_0 = 4\pi \times 10^{-7}$ H/m is the vacuum permeability and $\rho$ is the resistivity above $T_c$. Therefore, $\lambda_m(T)$ curves in Figure 3 change the behavior entering the normal state and above $T_c(H)$ follow the temperature dependent $\delta_{\text{skin}}(T)$. In fact, such measurements can be used for contactless resistivity measurements [75].

We now evaluate the Campbell length, $\lambda_C = \sqrt{\lambda_m^2 - \lambda_L^2}$. Figure 4 shows the calculated $\lambda_C(T)$ with the same type and color of the curves for the indicated magnetic fields as in Fig.3. With an increasing magnetic field, the $\lambda_C(T)$ curves move upward, indicating the field dependence, which can be determined from Fig.4 by taking isothermal slices. The result is shown in 5 where the top panel shows the ZFC $j_c(B)$ curves, while the bottom panel shows the FC curves.

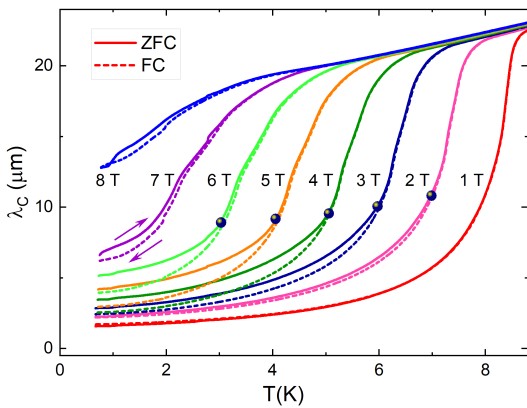

Figure 4: Campbell penetration depth, $\lambda_C = \sqrt{\lambda_m^2 - \lambda_L^2}$, as a function of temperature measured on warming (ZFC, solid lines) and cooling (FC, dashed lines) in different dc magnetic fields, shown next to each curve. The arrows indicate the direction on one of the curves. Symbols show the irreversibility temperature, $H_{irr}$.

In the original Campbell model [55,56], $\lambda_C^2 = \phi_0 H / \alpha$ and $j_c = \alpha r_p / \phi_0 = H r_p / \lambda_C^2$, where $r_p$ is the radius of the pinning potential, usually assumed to be approximately equal to the coherence length, $\xi$, and $\alpha$ is the so-called Labusch constant, the curvature of the pinning potential, $\alpha = d^2U/dr^2$. Note that we used SI units with $H$ being the magnetic field strength in A/m. When magnetic induction $B$ is used in tesla, the formulas must replace $H = B/\mu_0$.

Generally, the Campbell length $\lambda_C(H) \propto H^\beta$ with $\beta = 0.5$ in the original Campbell theory, which was observed in the systems previously studied [76,77]. However, other superconductors show a different exponent $\beta$. For example, a more concave $\lambda_C(H)$ with $\beta = 0.25$ was observed in the low carrier density superconductor YPtBi [78]. In the present case of $Mo_8Ga_{41}$, a very different behavior with a field-dependent exponent, $\beta(B) > 0.5$, indicating that the pinning potential is anharmonic with a field-dependent Labusch parameter. There is an important difference between the ZFC and FC protocols. While the former probes the Labusch parameter closer to the edge of the pinning potential where vortices are biased by the Bean persistent current, $j_p$, the FC measurements probe the theoretical critical current with vortices oscillating at the bottom of the pinning potential [79]. This is a good proxy for the true critical current density since the difference between the ZFC and FC branches is not large and the degree of anharmonicity of the pinning potential is small.

At low fields and temperatures, the FC curves, $\lambda_C(H)$, start with the exponent $\beta$ close to 0.5, but then change to a much larger value. This is likely due to a crossover from a single vortex pinning to the collective pinning of vortex bundles [44].

Figure 6 shows the theoretical critical current density evaluated from the Campbell model using the data from Fig. 4, $j_c = r_p H / \lambda_C^2$ [55, 56]. The range of the pinning potential, $r_p$, is usually associated with the coherence length, which does not change much over a large temperature interval. To be thorough, we extracted the temperature-dependent $\xi(T)$ from the experimental upper critical field [17], and found that it only doubles at $T = 7.5$ K. To facilitate the comparison and avoid overcrowding, Fig. 6 shows the theoretical critical current density as a function of temperature for two values of the applied magnetic field, 2 T and 4 T. Symbols show the results with a temperature-dependent $\xi(T)$, while dashed lines assume a fixed $r_p = \xi(0) = 6.4$ nm. Expectedly, the difference increases for larger temperatures, but it is still minor and does not change the overall functional form. In ZFC measurements, the curves at 2T lie below those of 4T, so that $j_c(2\,\text{T}) > j_c(4\,\text{T})$, showing an expected monotonic decrease of $j_c$ with a magnetic field. Surprisingly, the FC curves reverse this order below roughly 4 K, so $j_c(2\,\text{T}) < j_c(4\,\text{T})$, showing an increase of $j_c$ with increasing magnetic field.

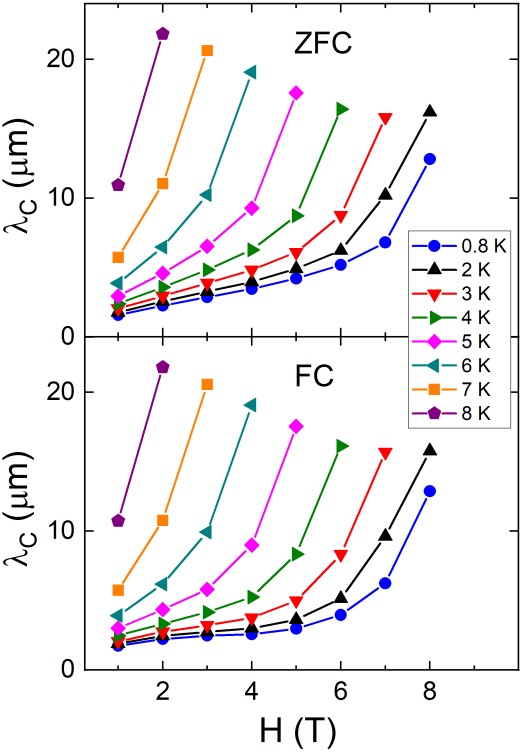

Figure 5: Campbell length, $\lambda_C$, as a function of a magnetic field at fixed temperatures as shown in the legend. The symbols mark the points obtained from the isothermal slices of Fig.4.

We now examine the magnetic field dependence of the theoretical $j_c(H)$ evaluated at several temperatures for both protocols. The upper panel of Fig.7 shows the ZFC data, while the bottom panel shows the FC measurements. As before, solid lines and symbols show $j_c(H)$ calculated with temperature-dependent $\xi(T)$, while dashed lines assume $r_p = \xi(0) = 6.4$ nm. The two estimates are close and do not change the general picture. For comparison, Fig.7 also shows the persistent current density extracted from conventional magnetization measurements at $T = 2$ K estimated from the Bean model [37]. The overall amplitude is quite comparable, but the actual field dependence is quite different, probably because of the effect of magnetic relaxation during the time window of the measurement.

The lower panel of Fig.7 presents an unusual result. There is a pronounced peak effect in $j_c(H)$ in the FC protocol when vortices oscillate at the bottom of their pinning potential wells in the absence of a vortex-biasing Bean persistent current. This information is inaccessible in conventional magnetization measurements, which are always accompanied by the vortex density gradient, and the measured signal is proportional to this gradient, hence the persistent current density $j_p(H)$. This result implies that the pinning potential changes non-monotonically with increasing magnetic field. Since $j_c = \alpha r_p / \phi_0$, this means that the product $\alpha r_p$ is non-monotonic. The Labusch constant does not depend on $r_p$, but depends on the depth of the effective pinning potential, $U_0$.

In principle, the observed peak effect in a FC protocol can be explained within the collective pinning theory [44], but suggesting a novel mechanism of its formation. As mentioned in the Introduction, there are two explanations of the peak effect: static when there is a nonmonotonic critical current $j_c(H)$, and dynamic in which the peak in the persistent current density is formed as a result of field-dependent magnetic relaxation, which is faster at lower magnetic fields [44]. Our results suggest the existence of the third scenario, which is essentially a com-

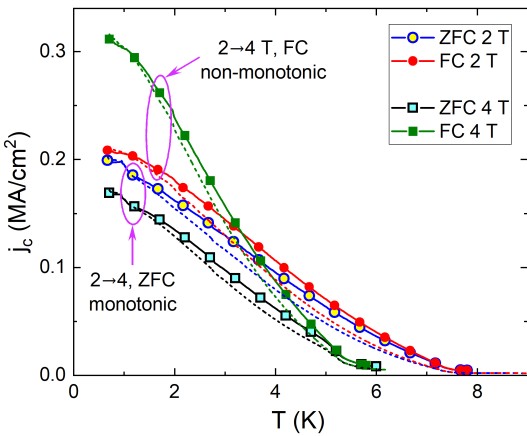

Figure 6: Temperature dependence of the theoretical critical current density in 2 T (circles, empty - ZFC, filled - FC) and 4 T (squares) evaluated from $j_c = H r_p / \lambda_C^2$ using the data shown in Fig.4. The dashed lines are calculated with a constant $r_p = \xi(0) = 6.4$ nm. Ellipses group the data into ZFC and FC protocols. The critical current decreases with an increasing magnetic field in the entire temperature interval in a ZFC protocol $j_c(2\,\mathrm{T}) > j_c(4\,\mathrm{T})$, but the trend is opposite in the case of a FC protocol below roughly 4 K.

bination of the two. The critical current is monotonic in the ZFC state with a vortex density gradient but is non-monotonic in the relaxed FC state without the gradient. This difference is possible for an anharmonic pining potential. However, to arrive at this nonmonotonic peak-effect state, the system needs to relax. The relaxation rate may be a monotonic function of a magnetic field or not.

Of course, a detailed microscopic explanation of the observed "hidden" peak effect is likely more complicated. After all, we used a simplified Campbell picture, which may not be quantitatively applicable here. However, we believe that even a simplified discussion captures the key aspects of the results.

Finally, we conclude by constructing the $H-T$ vortex phase diagram. Figure 8 shows the upper critical field extracted from the onset of $\lambda_m(T)$ curves shown in Fig.3. For comparison, $H_{c2}(T)$ estimates in the literature from the onset of magnetization are shown by black "+" symbols [16] and by green "x" symbols [17]. The irreversibility line, obtained as a point where the ZFC and FC curves separate, Fig.3, is shown by blue symbols. The vertical midpoint line of the $\lambda_m(T)$ dependencies (green squares) follows closely. This phase diagram shows that the effects of anharmonicity diminish at higher temperatures and larger magnetic fields.

## 4  Conclusions

In conclusion, London and Campbell penetration depths were systematically investigated in single crystals of the endohedral gallide cluster superconductor, $Mo_8Ga_{41}$. The full temperature range superfluid density is consistent with the clean isotropic $s$−wave weak-coupling BCS theory without any signs of the second gap or strong coupling. The critical current density evaluated from the Campbell length reveals an unusual result. Its field dependence is monotonic in the zero-field cooling (ZFC) process, but exhibits a profound "hidden" peak effect in the field-cooling (FC) protocol. It is hidden because there is no vortex density gradient in a FC protocol, whereas conventional measurements of the irreversible state are always accompanied by a vortex density gradient that supports the persistent Bean current.

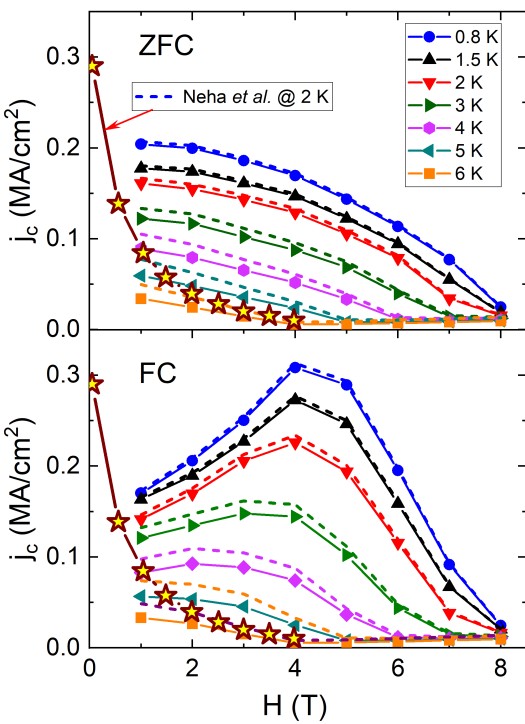

Figure 7: Campbell penetration depth, $\lambda_C$, as a function of an applied magnetic field at several fixed temperatures listed in the legend. The upper panel shows ZFC data, and the bottom panel shows FC results. Also shown is the persistent current density extracted from the conventional magnetization measurements at $T = 2\,\text{K}$ using the Bean model [37].

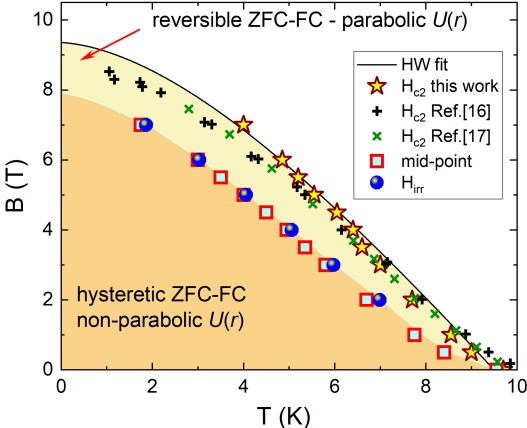

Figure 8: Vortex matter phase diagram constructed from $\lambda_m(T, H)$. The upper critical field, $H_{c2}(T)$, is estimated using the onset of the diamagnetic transition as a criterion (stars). For comparison, the literature $H_{c2}(T)$ estimated from the onset of magnetization is shown by black "+" symbols [16] and by green "x"symbols [17]. The irreversibility line below which the ZFC and FC Campbell lengths split is shown by blue circles. It is practically the same as the midpoint of the transition curves shown by squares.

We suggest that at least in some compounds, the peak effect appears as a result of magnetic relaxation (long time window of the experiment), but not because this relaxation is magnetic field dependent (which is still possible, though). Instead, the system evolves from an anharmonic regime with monotonic critical current, $j_c(H)$, to a relaxed harmonic regime at smaller persistent current amplitude, $j_p(H) \ll j_c(H)$, where the vortex potential itself is a non-monotonic function of a magnetic field. Therefore, this scenario of peak effect formation involves both static and dynamic aspects.

## Acknowledgments

We thank V. Geshkenbein for the useful discussions.

**Funding information** This work was supported by the US DOE, Office of Science, BES Materials Science and Engineering Division under contract # DE-AC02-07CH11358. Work at BNL (materials synthesis) was supported by the U.S. Department of Energy, Basic Energy Sciences, Division of Materials Science and Engineering, under Contract No. DE-SC0012704. C.P. acknowledges support from the Shanghai Key Laboratory of Material Frontiers Research in Extreme Environments, China (No. 22dz2260800) and Shanghai Science and Technology Committee, China (No. 22JC1410300).

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
