# Peer review of "Conventional single-gap $s-$wave superconductivity and hidden peak effect in single crystals of Mo$_{8}$Ga$_{41}$ superconductor"

_SciPost Physics, doi:SciPost Phys. 17, 117 (2024)_

## Round 1 · Referee Report · Anonymous (Referee 1) · 2024-7-31

Strengths
Weaknesses
Report
Requested changes
A) Technical remarks, I would recomend to:
A1 - use proper expression for the ratio of the energy gap and the critical temperature (involve Boltzmann constant), especially when you refer to specific numbers.
A2 - try to avoid mixing SI and CGS units in the same text. CGS is obsolete - it is easy to use mT instead of Oe.
A3 - in some parts of the text there are references to numbers of the sections (Section II, Section III.B) even if the sections are not numbered. It seems that it is some residuum from an older manuscript layout. Specially refering to Section III.B in a third section and second subsection is confusing.
B) The introductory part seems to be confused:
B1 - The authors claim that in Ref.9 the larger gap gives the ratio Delta/k_BT_c =1.857. However, this was calculated considering the smallest value from the distribution of the larger gaps (1.45 meV), moreover, local T_c of the junction with this energy gap was not determined. On the contrary, the temperature dependence of the two energy gaps with clearly identified critical temperature presented in Ref.9 gives the value of Delta 1.6, that leads to the ratio of 2.06, which is larger than the weak-coupling limit.
B2 - The authors claim that in Ref.16 the larger from the two gaps has the ratio even smaller than the weak-coupling limit. This is not true. Actually it is around 5, which is really unexpectedly far above the weak-coupling limit.
B3 - The authors comment on the results presented in Ref.17, stating that the data can be fit well both with single-gap and two-gap fit with minor differences. This is true, but the main message of the paper is that none of the options is valid, because the single-gap fit leads to the ratio much lower than the weak-coupling limit and the two-gap fit is not consistent with the heat capacity measurement performed on the same crystal. Instead, the paper is a warning that the measurement of the superfluid density itself, without knowing the complex picture, might be misinterpreted e.g. for a two-gap feature. This context is not mentioned in the Introduction.
C) In the section Samples and methods, subsection Lower critical field, the authors claim that the measurements are performed as close to the sample edge as possible and refer also to Ref.17. However, in Ref.17 it is explained that for the measurement of the penetration field in this sample featuring strong pinning (V-shaped profile), a probe is selected that is distant from the edge by the half-width of the sample. This does not mean "as close as possible", actually the selected probe was the forth in a row from the edge (see Fig.1 in Ref.17). I would suggest to elaborate this part - mentioning the sample shape and sizes, and estimate the position of the measurement in respect to the sample edge. Was some other position examined as well?
D) I wonder why the authors used weak-coupling limit for calculation of lambda. The other literature refers mostly to moderate or strong coupling (Ref.9: Delta/k_BT_c =2.06; Ref.15: jump in Cp 2.83>1.43; Ref.16: Delta/k_BT_c =2.1 if single-gap fit is considered; Ref.17/18: Delta/k_BT_c =2.2). What would be the difference in the penetration depth and subsequently in the superfluid density (since one needs to feed the value of lambda(0) for the calculation), if larger coupling was involved?
E) In the section Results and Discussion, subsection London penetration depth..., there is a sentence I consider to be too simplistic: "There is no indication of a multi-gap behavior, which usually appears as a convex curvature ..." The shape of the superfluid density depends on various parameters - mostly on how far are the two energy gaps apart and what is the weight of their contribution. It is true, that most of the clearly identified two-gap superconductors have this feature, but it is not general. In other words, absence of the positive curvature does not necessarily exclude two-gap superconductivity.
F) Decription of Fig. 3 looks confusing - the Y-axis is called lambda_m, in the figure caption it says that it shows Lambda_m(T)*lambda(0)+DeltaLambda(T) and in the text it states "note that the measured DeltaLambda_m(T)..." Also the notion of saturation is rather confusing, since all the curves in Fig.3 increase at elevated temperatures, while their values overlap.
G) The first sentence of the section Results..., subsection Campbell penetration depth... looks confused, probably with wrong position of the bracket.
Recommendation
Ask for minor revision
Author: Ruslan Prozorov on 2024-08-31 [id 4727]
(in reply to Report 1 on 2024-07-31)In the attached PDF, we provide a detailed response to both referees. Our responses are in red, and changed text is in blue.
Attachment:
Referee_comments_Reply_Mo8Ga41_31-Aug-2024.pdf

---

## Round 1 · Referee Report · Anonymous (Referee 2) · 2024-8-28

Report
Mo8Ga41 is a potentially unconventional superconductor that has been controversially discussed in the previous literature. The manuscript under consideration reports the very first penetration depth measurement that demonstrates an unexpected but remarkably good agreement with the BCS expression in the clean limit. This interesting result sheds new light on the physics of Mo8Ga41 and potentially meets the SciPost Physics criterion of presenting a breakthrough in an existing research direction. Moreover, a curious peak effect in the critical current is reported and analyzed. I believe that this manuscript could be published in SciPost Physics, but it requires a major revision along the following lines:
-
One major puzzle is the simple BCS behavior of the superfluid density, as opposed to several previous observations of the "less conventional" superconductivity in Mo8Ga41. Some of these reports could be affected by the ambiguous fitting of the muon data or by surface effects, but bulk probes such as heat capacity should be more tenable. The large jump in the specific heat at Tc [PRB'2016] does indicate some physics beyond the weak-coupling limit of BCS. This observation directly contradicts the conventional scenario advocated by the present work. A plausible explanation for this discrepancy should be given.
-
One plausible explanation is the sample dependence. In this regard, details of the sample characterization are essentially missing, and it is not even clear whether exactly the same crystal(s) as in Ref. [14] have been used. I strongly encourage the authors to provide additional characterization data, such as:
- resistivity
- magnetization
-
heat capacity that could be compared to the previous publications. It would also be natural to supply Fig. 8 with the Bc2 values from the existing literature. Do they match?
-
Another possibility is that the superconductivity of Mo8Ga41 is not of the simple BCS type, but a more complex scenario mimics the BCS temperature dependence of the superfluid density, such that the good match in Fig. 2 is merely accidental. This idea may not be so odd if the physics of the normal state is considered. Mo8Ga41 is far from being a simple metal. Its resistivity exceeds the MIR limit and shows a rather peculiar temperature dependence below 100 K. The mean-free path should be quite short. Would one really expect the clean limit to be applicable in this case? Should not the dirty limit be more appropriate? What is the mean-free path in your sample?
-
The lower critical field was determined by the NV-center magnetometry. The Bc1 value at 4.25 K is somewhat lower than reported in PRB'2016: 85 Oe vs. 110-115 Oe. What is the reason for this difference? How was the sample surface prepared?
-
On page 5 there is a typo in the description of the critical current at 2 T and 4 T in the FC and ZFC regimes. The sentence with "the ZFC curves reverse this order" should probably read as "the FC curves reverse this order". According to Fig. 6, the order is reversed below 4 K only. Why does it not happen at higher temperatures? Would it be possible to shed more light on the vortex state of Mo8Ga41? For example, is there a vortex liquid, and in which temperature range?
Recommendation
Ask for major revision

---

## Round 2 · Referee Report · Anonymous (Referee 1) · 2024-9-13

Report
Dear Editor, Authors, I want to acknowledge the effort the authors spent improving the manuscript. However, there are still some issues to be cleared.
1) In response to comment B3, the authors confused Ref.17 and Ref.16 of the original manuscript version. While the authors' reaction is related to the measurements of Verchenko et al. (2017 and 2016), my concern was related to the publications Marcin et al. (2021 and 2019). So once again: The authors comment on the results presented in Ref.18 (Marcin 2021, Ref.17 in the previous version of the manuscript), stating that the data can be fit well both with single-gap and two-gap fit with minor differences. This is true, but the main message of the paper is that none of the options is valid, because the single-gap fit leads to the ratio much lower than the weak-coupling limit and the two-gap fit is not consistent with the heat capacity measurement performed on the same crystal. Instead, the paper is a warning that the measurement of the superfluid density itself, without knowing the complex picture, might be misinterpreted e.g. for a two-gap feature. This context is not mentioned in the Introduction.
2) On top of that, I think that the newly presented discussion related to the results of Verchenko et al. (including the new reference, Carbotte 1990) is not helping to clear things. In the curves in Fig.110 of Carbotte1990, that could possibly be related to MoGa sample, there is no convex curvature at all. For the coupling around 4.4, T_c/omega_ln is around 0.12 (dotted curve in Fig.110, only slightly different from the weak coupling behavior). Even if we admit that the coupling ratio would be around 5, the T_c/omega_ln would be aroung 0.16 (see Fig.31 in the same document), which is then something between dotted and dashed curve in Fig.110, still not that different from BCS. Actually, the “convex” curvature occurs at much much higher coupling ratios.
3) I think it is important that the new paper diminishes confusion in the available data and presents them in context, not just providing a simple list of previous results. So, I suggest to reorganize the introductory part. I understand that the authors do not want to analyze published results of other groups, as mentioned in the reply related to STM measurements of Sirohi2019. On the other hand, the authors have the opportunity to see things after they evolved in time, and they can and should put them in context after the critical judgment. Verchenko in 2016, in their first report of superconductivity in MoGa claimed a very large jump in the heat capacity (2.83) and suggested strong coupling. However, if you inspect Fig.7 in Verchenko2016, you realize that the heat capacity in zero magnetic field goes below zero at low temperatures, pointing to some issues with the data (specifically, in this case, it is not correct then to derive gamma simply from the extrapolation of the 11T measurement, because you get underestimated value that will give you in the end overestimated value of the jump vs gamma). Later on, their heat capacity data appears once again in Marcin2019 in Fig.1 after (probably correct) subtraction of the normal state contribution, and the jump turns out to be only 1.9 (or 2 at best, if you take the alpha model presented in the figure). This casts doubts to their former interpretation of muSR data being related to two-gap superconductivity. Similarly, Sirohi2019 showed two-gap spectra in their STM measurements, but Marcin2019 brought an explanation that it is “just a happy case” when your tunneling current goes simultaneously to two grains (supported by the fact that single gap spectra were also measured, consistent with the heat capacity measurements, which is a bulk probe). Later Marcin2021 presented a message “be careful if you study results of only one experimental method, because it might lead to misinterpretation of the data” – see Conclusions in the mentioned paper. I think this context should be elaborated.
4) Comment D in the previous report considered the calculation of the zero-temperature penetration depth lambda(0). I think, from the reply, it was misunderstood. I do not question comparing the superfluid density with the model, instead, I question the procedure of getting the superfluid density data. In the TDR measurement, in order to obtain superfluid density, it is necessary to feed the value of lambda(0). This value quite strongly affects the overall shape of the resulting temperature dependence of the superfluid density (see e.g. Fig.3 in Diener et al., Phys. Rev. B 79, 220508(R) ). In other words, you need to know lambda(0) before you actually start fitting your superfluid density. My concern was related to this: if the sample features moderate coupling and you consider Hc1(T) being weak-coupling instead, this would lead to an overestimation of Hc1(0), since at moderate coupling the Hc1(T) curve flattens earlier with decreasing temperature. Subsequently, the value of lambda(0) would be underestimated. This (following Fig.3 in in Diener et al., Phys. Rev. B 79, 220508(R) ) could lead to temperature dependence of the superfluid density looking more like a weak-coupling type. Moreover, it is not true, that the heat capacity relies only on the jump at Tc – in Marcin2019, the overall temperature dependence of the heat capacity is compared to the model and it fits well. In the end, the question remains – how would the superfluid density look like if for the calculation of lambda(0) moderate coupling (slightly above 4) was considered?
5) There is still a Boltzmann constant missing in the Introduction. Regardless of what is common, I think it is kind to not confuse the readers, especially students or junior researchers.
Recommendation
Ask for minor revision

Ruslan Prozorov on 2024-09-01 [id 4728]
REVISED RESPONSE to comment B3 of Referee 1:
We found a reference where the effect of strong coupling on superfluid density is calculated and compared to weak-coupling BCS. This significantly strengthens our original arguments (which remain valid). Please consider this updated version of the response to Referee 1 Comment B3:
Reference 17 shows an excellent single gap fit, Fig.7, and the text below this figure says: “The fit using Eq. (4) is shown in Fig.7 as a dashed line. The single-gap BCS-type model satisfactorily describes the experimental data with Δ(0) = 1.80(7) meV, Δ(0) /Tc=2.1…”. This value is a bit larger than 1.76 of the clean limit but could be due to the fact that the authors used a dirty-limit expression for fitting.
However, as the Referee pointed out, the authors of Ref.17 attempt to reconcile their results with previous specific heat measurements by the same group and arrive at some very large gap values, far exceeding the weak coupling limit. However, their two-gap analysis is not self-consistent, and one can find many different pairs of gap values of different amplitudes to fit the data using this simplistic approach.
To understand the effect of strong coupling on the superfluid density, Eliashberg theory must be used. Even for a single s-wave gap, it predicts significant differences between weak-coupling BCS and the strong coupling, practically at any coupling strength, see Fig.110 in Ref. {Carbotte1990}. [J. P. Carbotte, Properties of boson-exchange superconductors. Rev. Mod. Phys. 62, 1027 (1990)]. Expectedly, the superfluid density curve becomes much more convex, consistent with the large gap to Tc ratio. Clearly, our data and the data of Ref.17 (uSR study) are very different from this predicted behavior.
We note that there are a few superconductors (e.g. Rh17S15) where a specific heat jump at Tc is much larger than expected from the weak-coupling theory, but it is unclear if this signifies a strong coupling or if there are other contributions to the free energy from other degrees of freedom.
The corresponding part of the introduction is rewritten:
"Specific heat measurements reported a normalized jump at the superconducting transition, $\Delta C/C(T_c)=2.83$, twice the value of 1.43 predicted by weak-coupling theory \cite{Verchenko2016}. However, muon spin resonance ($\mu\text{SR}$) measurements by the same group reported superfluid density consistent with a single-gap weak-coupling isotropic BCS theory \cite{Verchenko2017}. To reconcile with their specific heat study \cite{Verchenko2016}, the same data were fitted with two very large gaps, 4.3 meV and 1.76 meV, resulting in gap ratios, $\Delta(0)/T_c=$5.1 and 2.1, respectively. However, these calculations use a non-self-consistent $\alpha-$model-type empirical formulation with approximate BCS temperature dependencies for both gaps, and using a dirty limit BCS formula. The proper self-consistent strong coupling analysis using Eliashberg theory shows that even in the case of isotropic $s-$wave, the gap temperature dependence is very different from the weak-coupling BCS \cite{Marsiglio1991}, resulting in superfluid density, which is much more convex and steep at higher temperatures, see Fig.110 in Ref.\cite{Carbotte1990}. This superfluid density is very different from our results or the results of the $\mu$SR study \cite{Verchenko2017}, which are similar."

---

## Round 2 · Referee Report · Anonymous (Referee 2) · 2024-9-25

Report
I would like to thank the authors for providing additional clarifications in response to my comments. I generally agree with the authors' reasoning, but unfortunately not all the necessary information has found its way into the manuscript yet.
-
Please, write explicitly that the same crystals as in Ref. [14] have been used. The simple citation of Ref. [14] could mean that new crystals were prepared by the same method, but probably in a different lab or using different precursors, potentially resulting in a different behavior. This information is especially crucial in the light of surface phases and other complications reported in the previous Mo8Ga41 literature.
-
Please, provide additional evidence against sample dependence between your samples and the samples studied by other groups. This can be done, for example, by: i) adding the Hc2 values from Verchenko et al. (2016) to Fig. 8; ii) directly comparing the superfluid density with the muSR data (Verchenko et al. 2017) This comparison is already alluded to in the authors' response. I am sure that readers will also benefit from seeing it.
-
I agree that the observation of quantum oscillations gives strong evidence for the clean limit, but readers would certainly benefit from an explicit comment on the choice of the clean limit over the dirty limit. I also found an interesting publication, PRB 102, 241113(R) (2020), that analyzes T-linear resistivity of Mo8Ga41 and ascribes it to low-lying phonon modes. It is then more agreeable that Mo8Ga41 is in the clean limit, even though its resistivity is quite high. I believe that the aforementioned reference may be worth citing.
I would also like to mention that the other reviewer's comments offer a broader perspective on the previous results for Mo8Ga41. It would be good if this perspective appears in the manuscript instead of being hidden in the referee report.
Recommendation
Ask for minor revision

---

## Round 2 · Author Response

Warnings issued while processing user-supplied markup:
- Inconsistency: plain/Markdown and reStructuredText syntaxes are mixed. Markdown will be used.
Add "#coerce:reST" or "#coerce:plain" as the first line of your text to force reStructuredText or no markup.
You may also contact the helpdesk if the formatting is incorrect and you are unable to edit your text.
Dear Editors,
We thank the Referees for their detailed feedback, recommendations, and the time and effort they put into reviewing this manuscript. We are pleased that both Referees recommended publication in SciPost Physics based on the novelty and importance of the results after their comments were addressed. We answer each point below.
REFEREE 1
The manuscript comprises a comprehensive study of magnetic field penetration to a single-crystalline Mo_8Ga_41 superconductor. It brings well presented, original and interesting results. In the manuscript sometimes shortcuts are present, some parts are confusing. I would recommend the publication after the authors clarify/correct the issues listed below. RESPONSE: We thank the referee for this very positive comment and publication recommendation. A1) use proper expression for the ratio of the energy gap and the critical temperature (involve Boltzmann constant), especially when you refer to specific numbers.
RESPONSE: In the current literature on superconductivity, it is quite common to set Boltzmann’s constant to 1 because what matters is the dimensionless ratio of the gap amplitude to the thermal energy, kBT. However, we followed the Referee's recommendation and added the Boltzmann constant, kB, where needed. A2) try to avoid mixing SI and CGS units in the same text. CGS is obsolete - it is easy to use mT instead of Oe. RESPONSE: This is a typical case where the so-called “practical” system of units is used. Here, the applied field strength, H, is in oersted, and the magnetic induction, B, is in tesla. In SI, the former is measured in A/m, an inconvenient and rarely used unit. We could use all CGS units and express the B-field in gauss, but tesla is better for its factor of 10000. We note that our magnetometer software produces data in CGS units.
A3) In some parts of the text there are references to numbers of the sections (Section II, Section III.B) even if the sections are not numbered. It seems that it is some residuum from an older manuscript layout. Specially referring to Section III.B in a third section and second subsection is confusing.
RESPONSE: We have reviewed the proper numbering of sections and subsections, as suggested.
B1) The authors claim that in Ref.9 the larger gap gives the ratio Delta/k_BT_c =1.857. However, this was calculated considering the smallest value from the distribution of the larger gaps (1.45 meV), moreover, local T_c of the junction with this energy gap was not determined. On the contrary, the temperature dependence of the two energy gaps with clearly identified critical temperature presented in Ref.9 gives the value of Delta 1.6, that leads to the ratio of 2.06, which is larger than the weak-coupling limit.
RESPONSE: While the text of Ref.9 quotes 1.6 meV and 0.9 meV for the two gaps, and these values appear in the field-dependence of the gaps, Fig.2(d), the temperature dependence, Fig.2(b), shows the gaps of 1.46 meV and 0.9 meV at T→0. Furthermore, the text says (bottom left column of page 3): “For the larger gap of Mo8Ga41, 2\Delta/k_B Tc is found to be 3.5, which is close to the expected value for a weak-coupling BCS superconductor.” We also note that the two-gap fitting of the tunneling spectra involves a fairly large phenomenological Dynes parameter Γ, which signifies some issues with the tunneling barrier. It is known that larger Γ tends to push the gap peaks to larger values. Of course, this is not our place to analyze Ref.9 in detail. We therefore revised our text in the introduction to read:
A scanning tunneling microscopy (STM) study of Mo$_{8}$Ga$_{41}$ crystals suggests a weak-coupling superconductivity with two gaps of 1.6 meV and 0.9 meV, resulting the ratio, $\Delta(0)/T_c$, of 2.1 and 1.2, and concludes that this behavior is similar to MgB$_2$ \cite{Sirohi2019}.
B2) The authors claim that in Ref.16 the larger from the two gaps has the ratio even smaller than the weak-coupling limit. This is not true. Actually, it is around 5, which is really unexpectedly far above the weak-coupling limit.
RESPONSE: The Referee is correct. We now mention this in the revised and extended text, please see a combined response to B2 and B3.
B3 - The authors comment on the results presented in Ref.17, stating that the data can be fit well both with single-gap and two-gap fit with minor differences. This is true, but the main message of the paper is that none of the options is valid, because the single-gap fit leads to the ratio much lower than the weak-coupling limit and the two-gap fit is not consistent with the heat capacity measurement performed on the same crystal. Instead, the paper is a warning that the measurement of the superfluid density itself, without knowing the complex picture, might be misinterpreted e.g. for a two-gap feature. This context is not mentioned in the Introduction.
RESPONSE: Reference 17 shows an excellent single gap fit, Fig.7, and the text below says: “The fit using Eq. (4) is shown in Fig.7 as a dashed line. The single-gap BCS-type model satisfactorily describes the experimental data with Δ(0) = 1.80(7) meV, \Delta(0)/kBTc=2.1…”. This value is a bit larger than 1.76 of the clean limit but could be due to the fact that the authors used a dirty-limit expression for fitting. However, as the Referee pointed out, the authors of Ref.17 attempt to reconcile their results with previous specific heat measurements by the same group and arrive at some very large gap values, far exceeding the weak coupling limit. However, their two-gap analysis is not self-consistent, and one can find many different pairs of gaps of different amplitudes to fit the data. A full Eliashberg-type analysis deriving temperature-dependent gaps from the self-consistency equations and then evaluating the superfluid density has to be performed. Considering how much the gap(T) changes in the strong-coupled superconductor (new reference: Marsiglio and Carbotte 1991) it is certain that the superfluid density with such large gaps will look very different from the weak-coupling single-gap BCS.
We note that there are a few superconductors (e.g. Rh17S15) where a specific heat jump at Tc is much larger than expected from the weak-coupling theory, but it is unclear if this signifies a strong coupling or if there are other contributions to the free energy from other degrees of freedom. This part of the introduction is rewritten.
C) In the section Samples and methods, subsection Lower critical field, the authors claim that the measurements are performed as close to the sample edge as possible and refer also to Ref.17. However, in Ref.17 it is explained that for the measurement of the penetration field in this sample featuring strong pinning (V-shaped profile), a probe is selected that is distant from the edge by the half-width of the sample. This does not mean "as close as possible", actually the selected probe was the forth in a row from the edge (see Fig.1 in Ref.17). I would suggest to elaborate this part - mentioning the sample shape and sizes, and estimate the position of the measurement in respect to the sample edge. Was some other position examined as well?
RESPONSE: We used a cuboid-shaped sample with dimensions: 400 um 200 um190 um. In Ref.17, it is clear that there is a significant uncertainty in the actual sample edge location and the elevation of the Hall probe above the surface. If they could, they would choose the probe closest to the edge, but the first probe where flux penetration is clearly identified is probe #6, with probe #5 being the “apparent” edge. In our case of NV centers dispersed in a diamond film, we have a much finer spatial resolution. The spot where we could clearly identify the superconducting screening was 10 um into the sample. We updated the relevant description accordingly.
D) I wonder why the authors used weak-coupling limit for calculation of lambda. The other literature refers mostly to moderate or strong coupling (Ref.9: Delta/k_BT_c =2.06; Ref.15: jump in Cp 2.83>1.43; Ref.16: Delta/k_BT_c =2.1 if single-gap fit is considered; Ref.17/18: Delta/k_BT_c =2.2). What would be the difference in the penetration depth and subsequently in the superfluid density (since one needs to feed the value of lambda(0) for the calculation), if larger coupling was involved?
RESPONSE: This is the central point of our paper. The stronger coupling values, quoted in the comment, would result in the superfluid density significantly different from the weak coupling. The gap(T) changes dramatically – we now cite a paper by Marsiglio and Carbotte (1991). The gap value affects the superfluid density curve exponentially, so it is very sensitive to it. It is important that we fit the data in the full temperature range, whereas specific heat relies on the jump at Tc. On the other hand, the value of the London penetration depth is only in the pre-factor of the low-T expansion, and it is obtained from an independent measurement. Furthermore, we must point out that, despite of what is often done in the literature – using the gap/Tc ratio as a variable, one cannot use the weak-coupling BCS formulas to compute the superfluid density with arbitrary not-self-consistent gaps. A full Eliashberg calculation must be performed. Since we successfully fit the data with the simplest weak-coupling theory, we do not need to consider a stronger coupling because we know that the curve will certainly be different.
E) In the section Results and Discussion, subsection London penetration depth..., there is a sentence I consider to be too simplistic: "There is no indication of a multi-gap behavior, which usually appears as a convex curvature ..." The shape of the superfluid density depends on various parameters - mostly on how far are the two energy gaps apart and what is the weight of their contribution. It is true, that most of the clearly identified two-gap superconductors have this feature, but it is not general. In other words, absence of the positive curvature does not necessarily exclude two-gap superconductivity.
RESPONSE: This is correct. To address this verbatim, we introduced a new paragraph.
F) Decription of Fig. 3 looks confusing - the Y-axis is called lambda_m, in the figure caption it says that it shows Lambda_m(T)*lambda(0)+DeltaLambda(T) and in the text it states "note that the measured DeltaLambda_m(T)..." Also the notion of saturation is rather confusing, since all the curves in Fig.3 increase at elevated temperatures, while their values overlap.
RESPONSE: We have corrected the description as Lambda_m(T)= lambda(0)+DeltaLambda(T) and text as Lambda_m(T) instead of DeltaLambda_m(T). We re-phrased the description and, instead of “saturation” use “does not diverge”. Specifically, the text now reads:
Note that the measured $\lambda_{m}(T)$ does not diverge approaching the normal state. Of course, the actual penetration depth diverges at $T\rightarrow T_{c}\left(H\right)$, but in the normal state, the measured depth $\lambda_{m}$ cannot exceed the skin depth, $\delta_{\text{skin}}=\sqrt{\rho/\mu_{0}\pi f}$, where $\mu_{0}=4\pi\times10^{-7},\text{H/m}$ is the vacuum permeability and $\rho$ is the resistivity above $T_c$. Therefore, the $\lambda_{m}(T)$ curves in Figure~\ref{fig:Lm} change the behavior entering the normal state and above $T_{c}\left(H\right)$ follow the temperature dependent $\delta_{\text{skin}} (T)$. In fact, such measurements can be used for contactless resistivity measurements \cite{Prozorov2007}.
G) The first sentence of the section Results..., subsection Campbell penetration depth... looks confused, probably with wrong position of the bracket.
RESPONSE: It was corrected.
Referee 2
Mo8Ga41 is a potentially unconventional superconductor that has been controversially discussed in the previous literature. The manuscript under consideration reports the very first penetration depth measurement that demonstrates an unexpected but remarkably good agreement with the BCS expression in the clean limit. This interesting result sheds new light on the physics of Mo8Ga41 and potentially meets the SciPost Physics criterion of presenting a breakthrough in an existing research direction. Moreover, a curious peak effect in the critical current is reported and analyzed. I believe that this manuscript could be published in SciPost Physics, but it requires a major revision along the following lines.
RESPONSE: We thank the referee for this very positive comment and overall recommendation to publish this paper in SciPost Physics.
1) One major puzzle is the simple BCS behavior of the superfluid density, as opposed to several previous observations of the "less conventional" superconductivity in Mo8Ga41. Some of these reports could be affected by the ambiguous fitting of the muon data or by surface effects, but bulk probes such as heat capacity should be more tenable. The large jump in the specific heat at Tc [PRB'2016] indicates some physics beyond BCS's weak-coupling limit. This observation directly contradicts the conventional scenario advocated by the present work. A plausible explanation for this discrepancy should be given.
RESPONSE: This is a question similar to B3 of the Referee 1. We note that there are a few superconductors (e.g. Rh17S15) where specific heat jump at Tc is much larger than expected from the weak-coupling theory. It is unclear if this signifies a strong coupling or not. Perhaps, there are other contributions to the free energy (entropy) from other degrees of freedom. However, such large gap taken at a face value is simply incompatible with the measurements of superfluid density or tunneling. A full Eliashberg-type analysis of the superfluid density and temperature-dependent specific heat (not just its jump at Tc) is needed to address this important question.
2) One plausible explanation is the sample dependence. In this regard, details of the sample characterization are essentially missing, and it is not even clear whether exactly the same crystal(s) as in Ref. [14] have been used. I strongly encourage the authors to provide additional characterization data, such as resistivity, magnetization, heat capacity that could be compared to the previous publications. It would also be natural to supply Fig. 8 with the Bc2 values from the existing literature. Do they match?
RESPONSE: The samples used in this study were from the same batch as used described in Ref.[14], as can be seen from the shared co-authors between two works. No variation of properties was found within the batch. The resistivity and magnetization measurements were reported in the original paper, Ref.[14].
3) Another possibility is that the superconductivity of Mo8Ga41 is not of the simple BCS type, but a more complex scenario mimics the BCS temperature dependence of the superfluid density, such that the good match in Fig. 2 is merely accidental. This idea may not be so odd if the physics of the normal state is considered. Mo8Ga41 is far from being a simple metal. Its resistivity exceeds the MIR limit and shows a rather peculiar temperature dependence below 100 K. The mean-free path should be quite short. Would one really expect the clean limit to be applicable in this case? Should not the dirty limit be more appropriate? What is the mean-free path in your sample?
RESPONSE: It is not easy to estimate the mean mean free path in this multiband system. However, the RRR=17 is quite good. We also see from our Fig.2 that the dirty limit would result in distinctly different superfluid density. Finally, Ref.[17] reports well-resolved quantum oscillations, which is impossible in a dirty system.
4) The lower critical field was determined by the NV-center magnetometry. The Bc1 value at 4.25 K is somewhat lower than reported in PRB'2016: 85 Oe vs. 110-115 Oe. What is the reason for this difference? How was the sample surface prepared?
RESPONSE: Our NV center magnetometry is a local probe that looks at the field of first penetration. The magnetization measurement, such as in PRB 2016, is a bulk probe, and there is a significant issue of an unknown effective demagnetizing factor of a non-ellipsoidal sample. Furthermore, it is extremely hard to distinguish the moment when flux starts to penetrate the sample. All this results in a large uncertainty.
We screened numerous samples to identify the best one with well-defined edges. It's important to note that the same sample was utilized for both NV magnetometry and tunnel diode oscillator measurements to ensure consistency.
5) On page 5 there is a typo in the description of the critical current at 2 T and 4 T in the FC and ZFC regimes. The sentence with "the ZFC curves reverse this order" should probably read as "the FC curves reverse this order". According to Fig. 6, the order is reversed below 4 K only. Why does it not happen at higher temperatures? Would it be possible to shed more light on the vortex state of Mo8Ga41? For example, is there a vortex liquid, and in which temperature range?
RESPONSE: We have corrected the typo.
The Referee is correct. We clarified the statement and wrote:
Surprisingly, the FC curves reverse this order below roughly 4 K, so $j_{c}\left(2\:\text{T}\right)<j_{c}\left(4\:\text{T}\right)$, showing an increase of $j_{c}$ with increasing magnetic field.
RESPONSE: Regarding the vortex state studies. Indeed, this system shows interesting behavior. We plan to conduct ac and dc magnetic measurements and measure magnetoresistance to fully investigate vortex physics in this fascinating superconductor. Of course, this will take some time to complete.

---

## Round 2 · List of Changes

0) Numerous stylistic and grammar changes were made throughout the text.
Some of the major changes are:
1) The introduction was revised and partially rewritten.
2) We have added sample details as: “The dimensions of the sample used in this experiment was 400 × 200 × 190 µm and its effective is R = 44.53 µm, determined from the established calibration procedure..”
3) We introduced a new paragraph: “Addressing the possibility of two distinct gaps, we note that, unlike the not-self-consistent $\alpha-$model where temperature dependence of the two gaps is assumed to be BCS-like, the self-consistent two-band treatment shows that the smaller gap becomes significantly different from what is expected from the BCS \cite{Prozorov2011}….”
4) We have re-phrased the description in the Results section as: “Note that the measured $\lambda_{m}(T)$ does not diverge approaching the normal state. Of course, the actual penetration depth diverges at $T\rightarrow T_{c}\left(H\right)$, but in the normal state, the measured depth $\lambda_{m}$ cannot exceed the skin depth, $\delta_{\text{skin}}=\sqrt{\rho/\mu_{0}\pi f}$, where $\mu_{0}=4\pi\times10^{-7},\text{H/m}$ is the vacuum permeability and $\rho$ is the resistivity above $T_c$. Therefore, $\lambda_{m}(T)$ curves in Figure~\ref{fig:Lm} change the behavior entering the normal state and above $T_{c}\left(H\right)$ follow the temperature dependent $\delta_{\text{skin}} (T)$. In fact, such measurements can be used for contactless resistivity measurements \cite{Prozorov2007}.
5) We have re-phrased the description in the Results section as:
“Surprisingly, the FC curves reverse this order below roughly 4 K, so $j_{c}\left(2\:\text{T}\right)<j_{c}\left(4\:\text{T}\right)$, showing an increase of $j_{c}$ with increasing magnetic field.

---

## Round 3 · Author Response

Dear Editors, We appreciate the thoughtful and detailed comments from the referees. Here, we provide point-by-point answers and indicate the changes in the manuscript. We follow the numbering of the REPORTS.
REPORT 1
1) In response to comment B3, …..
Yes, there is a significant disagreement between the superfluid density and the specific heat, both in Verchenko 2017 and in Marcin 2021. We note that the superfluid density is a quantity directly connected with the superconducting gap structure, whereas total specific heat has multiple contributions from all possible excitations in the complex system. The electronic part is notoriously difficult to separate. Typically used quenching superconductivity by a magnetic field is problematic due to magnetoresistance in this system [new reference: Zhao 2020 in the revised manuscript]. The unusual specific heat is an interesting problem, but it is beyond the scope of the present work. Furthermore, the Referee’s own excellent literature analysis in section #3 lists several issues, both technical and fundamental. We agree with that statement.
We followed the advice of the Referee and reorganized and rewritten the introduction to discuss these issues, highlighting the work of Verchenko et al. and Marcin et al. In light of this discussion, we believe that our work contributes significantly to the unusual physics of the Mo8Ga41 superconductor.
2) On top of that, I think ...
One has to focus on the superfluid density obtained using London penetration depth, which is shown in Fig.111 of Carbotte’s paper, Ref. [24] of the revised manuscript. The Referee is correct that for a gap to Tc ratio of 2.2, the strong-coupling parameter Tc/omega_ln would be around 0.1, and the resulting superfluid density will be close to the dotted curve for 0.15 in Ref.[24]. In our view, the difference between this curve and the weak-coupling BCS is significant and would easily be observed in our measurements. Furthermore, the suggested two-band scenario has to be solved self-consistently within Eliashberg theory and not using an unphysical alpha-model. We address this in the updated introduction.
3) I think it is important...
We agree with this analysis, which nicely connects several works. We have thoroughly rewritten this part of the Introduction, focusing on the superfluid density vs. specific heat reported in the same papers, and discussed the shortcomings of the models used for the analysis. We have also addressed the important question of clean vs. dirty limit.
4) Comment D in the previous report …
In the clean limit, \lambda(0) is a normal state property and does not depend on any superconducting parameters. In the presence of scattering, it is renormalized but only weakly. According to Tinkham, \lambda(0) = \lambda(0)_clean*\sqrt(1+\xi_0/\ell) where xi_0 is the BCS coherence length and \ell is the mean free path, so the correction is relatively weak. It is more important, though, that \lambda(0) in our paper was obtained from direct and independent measurements using NV magnetometry in a model-independent manner. Therefore, our superfluid density is obtained from the experiment without any assumptions, and it is fully consistent with the weak-coupling BCS.
5) There is still a Boltzmann constant ...
We are sorry that we missed it in two places. It is corrected.
REPORT 2
- Please, write explicitly ...
In the original text we describe how the crystals used in our study were obtained, summarizing the protocol and giving the reference: “\textbf{Samples:} Single crystals of Mo$_{8}$Ga$_{41}$ were grown by the high-temperature self-flux method. Mo and Ga were mixed in an 8: 500 ratio in an alumina crucible and sealed in an evacuated quartz tube. The ampule was heated up to 850$^{0}$C in two hours, held at 850$^{0}$C for 10 hours, and then slowly cooled to 170$^{0}$C for 55 hours, when the crystals were decanted \cite{Petrovic_QO_2020}.” For added specificity and to address the Referee’s concern, we now added at the end of this paragraph: “The full details of crystal growth and characterization are given in Ref.\cite{Petrovic_QO_2020}.”
- Please, provide additional evidence ...
As the Referee pointed out, our superfluid density is very similar to the muSR data, since both can be fitted well by the weak-coupling BCS formula. Regarding the Hc2, we followed the Referee’s advice and now included the data not only from Verchenko 2016, but also from Marcin 2019. Both agree with our Hc2(T) data. Figure 8 is modified, and the text includes proper references.
- I agree that the observation of ...
We found additional information about the residual resistivity ratio and now write regarding the clean limit: “…Mo$_{8}$Ga$_{41}$ appears to be rather clean, as is evident from the residual resistivity ratio of 15.4 \cite{Marcin2019} and, importantly, from the observation of quantum oscillations \cite{Petrovic_QO_2020}. (Our data are also consistent with the clean limit).”
and in the discussion:
“As mentioned in the Introduction, the clean limit is independently supported by observations of quantum oscillations in Mo$_{8}$Ga$_{41}$ crystals \cite{Petrovic_QO_2020} and a not too low residual resistivity ratio of 15.4 \cite{Marcin2019}.” We thank the Referee for pointing out this interesting reference. We now cite it in the paper in the introduction of the revised text: “A quantum oscillations study of Mo$_{8}$Ga$_{41}$ inferred three-dimensional electronic bands with strong coupling to phonons \cite{Petrovic_QO_2020}. Similarly, a $T-$linear resistivity observed in a wide temperature interval was interpreted to be due to scattering from low-lying phonon modes, which could also lead to enhanced electron-phonon coupling \cite{Zhang2020}.
REPORT: I would also like to mention that the other reviewer's comments ...
Please, see our reply to REPORT 1, #3 above. We reorganized and rewritten the introduction by including the suggested discussion.

---

## Round 3 · List of Changes

Major changes:
The introduction was thoroughly revised following the points raised in the REPORTS.
Figure 8 was updated to include a comparison with literature Hc2.
The sample synthesis part is updated.
Introduced eleven new references.

---

## Editorial Decision

published